# A Comparative Study on CO_2_-Switchable Foams Stabilized by C_22_- or C_18_-Tailed Tertiary Amines

**DOI:** 10.3390/molecules28062567

**Published:** 2023-03-11

**Authors:** Meiqing Liang, Xuezhi Zhao, Ji Wang, Yujun Feng

**Affiliations:** 1Chengdu Institute of Organic Chemistry, Chinese Academy of Sciences, Chengdu 610041, China; liangmeiqinggucas@126.com; 2State Key Laboratory of Polymer Materials Engineering, Polymer Research Institute, Sichuan University, Chengdu 610065, China; zhaoxz@scu.edu.cn; 3Tianfu Yongxing Laboratory, Chengdu 610217, China

**Keywords:** CO_2_ aqueous foams, foam properties, ultra-long chain surfactants, CO_2_-switchable, surface tension

## Abstract

The CO_2_ aqueous foams stabilized by bioresource-derived ultra-long chain surfactants have demonstrated considerable promising application potential owing to their remarkable longevity. Nevertheless, existing research is still inadequate to establish the relationships among surfactant architecture, environmental factors, and foam properties. Herein, two cases of ultra-long chain tertiary amines with different tail lengths, *N*-erucamidopropyl-*N,N*-dimethylamine (UC_22_AMPM) and *N*-oleicamidopropyl-*N,N*-dimethylamine (UC_18_AMPM), were employed to fabricate CO_2_ foams. The effect of temperature, pressure and salinity on the properties of two foam systems (i.e., foamability and foam stability) was compared using a high-temperature, high-pressure visualization foam meter. The continuous phase viscosity and liquid content for both samples were characterized using rheometry and FoamScan. The results showed that the increased concentrations or pressure enhanced the properties of both foam samples, but the increased scope for UC_22_AMPM was more pronounced. By contrast, the foam stability for both cases was impaired with increasing salinity or temperature, but the UC_18_AMPM sample is more sensitive to temperature and salinity, indicating the salt and temperature resistance of UC_18_AMPM-CO_2_ foams is weaker than those of the UC_22_AMPM counterpart. These differences are associated with the longer hydrophobic chain of UC_22_AMPM, which imparts a higher viscosity and lower surface tension to foams, resisting the adverse effects of temperature and salinity.

## 1. Introduction

Carbon dioxide (CO_2_) aqueous foams are colloidal dispersions composed of CO_2_ bubbles dispersed in a continuous aqueous phase [1]. Due to their relatively low density, larger surface area and excellent fluidity, CO_2_ aqueous foams have been widely used in many industrial processes and applications including the petroleum industry [2], ore flotation [3] and firefights [4]. The traditional CO_2_ aqueous foams were obtained using anionic surfactants such as sodium dodecyl sulfate (SDS) [5] and alpha olefin sulfonate (AOS) [6] as foaming agents by decreasing the CO_2_–water interfacial tension (C–W IFT) and capillary forces (P_c_). Unfortunately, such CO_2_ aqueous foams rapidly destabilized through a combination of drainage [7,8], coalescence [7,8], and Ostwald ripening [9]. As a result, the lifetime CO_2_ aqueous foams made of common surfactants do not exceed a few tens of minutes [10,11] and fail to satisfy the practical requirements. To improve foam lifetime, various foam stabilizers, including polymer [12,13], protein [14,15,16], and nanoparticles [17,18], are introduced into the aqueous foam systems against foam destabilization. In some cases, there is a demand for both the stable foam formed and controlled foam destruction. Taking cleaning processes as an example, the stable foam needs to be destabilized rapidly in a controlled way at the end of cleaning to obtain only a small volume of contaminated liquid that is easier to handle compared with foam [19]. Therefore, switchable or stimuli-responsive foams with tunable stability have been paid much attention in recent years.

During the past decade, the use of bioresource-derived ultra-long-chain surfactants (the hydrophobic chains ≥ C_18_) as stabilizers to prepare long-lasting aqueous foams has been widely reported and attracted significant interest. Johnston et al. [20] pioneered the utilization of erucylamidopropyldimethyl betaine (EAPB) to prepare long-lived CO_2_ foams. The CO_2_ foams stabilized by EAPB are intact at temperatures up to 120 °C and CO_2_ volumetric fractions up to 0.98. Likewise, our laboratory used *N*-erucamidopropyl-*N,N*-dimethylamine (UC_22_AMPM) to develop a CO_2_ aqueous foam with a lifetime of up to 6 h in 120 °C and 10 MPa [21]. The mechanism behind the foams stabilization by ultra-long-chain surfactants can be summarized as follows: (i) ultra-long-chain surfactants adsorb at CO_2_–water interfaces to form a dense surfactants layer in foam film, resisting coarsening and coalescence of bubbles [22]; (ii) the ultra-long-chain surfactants can assemble into viscoelastic aggregates, enhancing the solution viscosity and thus suppressing the liquid drains within the foam film [23,24]. In addition to their excellent foam stabilization capability, another merit of ultra-long-chain surfactants over petrochemical-based short-chain surfactants is being environmentally benign and sustainable as their feedstocks are natural renewable materials such as vegetable oil [25,26]. To provide scientific guidance for the application of CO_2_ foams in harsh conditions, previous research on ultra-long-chain surfactants stabilizing CO_2_ aqueous foams has been focused on establishing the relationships between various factors (e.g., pH, temperature, salinity and pressure) and foam properties [21,27]. However, these studies have typically been conducted in a single foam system, mainly rooted in the previous view that foam destabilization depends more on the mesoscopic properties of the foam such as bubble radius, foam film thickness and liquid fraction than on the chemical properties of the surfactant. Consequently, there are still insufficient insights into the contribution of surfactant structure to foam stability and evolution, impeding the advancement and exploitation of such foam systems.

In fact, many studies have demonstrated that surfactant structure has a noticeable impact on foam evolution and stability [16,28,29]. Fameau et al. [30] explored the role of tail length and head groups in foam properties by comparing the performance of foams made from long-chain fatty acids (myristic acid, palmitostearic acid, juniperic acid and 12-hydroxystearic acid). They found the foamability of fatty acid-based foam increased with decreasing the alkyl chain length of the fatty acid. Moreover, the presence of a hydroxyl group on the hydrophobic tail of the fatty acids increases the foamability in comparison to the non-hydroxylated fatty acids analog. A systematic study of foams made with a series of multi-tailed surfactants reported by Feitosa and co-workers demonstrated foams made with tri-cephalic double-tailed molecules have better stability than the single-tail one, regardless of the head structure [10]. Enlightened by these findings, we can safely hypothesize that aqueous foams stabilized by ultra-long-chain surfactants with different structures will exhibit differentiated foam properties and aging processes. In this context, it is desirable and beneficial to establish the correlation among the molecular structure of long-tailed surfactants, foam properties and evolution.

The objective of this study is to establish the surfactant structure-foam properties-foam evolution links and to deepen the understanding of the role of surfactant structure in foam properties. To attain this goal, UC_22_AMPM and its analog (UC_18_AMPM, C_18_ tail, Figure 1) were used as model compounds to develop CO_2_ aqueous foams. Then, the foaming ability and foam stability of two CO_2_ aqueous foam systems were meticulously compared at various temperatures, pressures and salinity using FoamScan and a high-temperature, high-pressure (HTHP) visualization foam meter. Meanwhile, the as-prepared CO_2_ aqueous foams were investigated by rheometer to unravel the underlying principles driving the discrepancies in foam properties.

## 2. Results and Discussion

The section is organized as follows: First, the concentration of UC_22_AMPM and UC_18_AMPM is optimized by the static foam test in atmospheric pressure at 35 °C. Then, the switching behavior of UC_18_AMPM-CO_2_ aqueous foam is characterized in comparison with UC_22_AMPM-CO_2_ aqueous foam. Finally, the influence of temperature, salinity, and pressure on the performance of CO_2_ aqueous foams UC_22_AMPM and UC_18_AMPM is examined, respectively.

### 2.1. Determination of Optimum Concentration

It is known that foam properties strongly depend on the concentration of the foaming agent [31,32]. Generally, the foaming properties are referred to as foamability (the maximum volume of foam system for a certain volume of foaming agent solution after a certain time of shear effect at a certain temperature, V_max_) [33,34] and foam stability (the time taken by the volume of foam system from V_max_ to a half at a certain temperature, t_1/2_) [33,34]. To determine the optimum concentration, the properties of UC_22_AMPM and UC_18_AMPM foams were investigated separately as a function of the concentration (0.1–0.5%) using FoamScan under atmospheric pressure at 35 °C. We previously demonstrated that UC_22_AMPM could form stable CO_2_ aqueous foams but not N_2_ ones [21,35]. Thus, CO_2_ was employed as the foaming gas in this work.

Figure 1A,B present the changes in V_max_ and t_1/2_ of both aqueous foams with increasing concentration, respectively. It can be seen the V_max_ for UC_18_AMPM was always constant at around 180 mL with increasing UC_18_AMPM concentration (*C*_UC18AMPM_), while the V_max_ of UC_22_AMPM rose from 179 to 189 mL (Figure 1A). Meanwhile, the t_1/2_ of both aqueous foams rose as the concentration increased (Figure 1B). From Table 1, the t_1/2_ of UC_22_AMPM-CO_2_ aqueous foams improved by 2.6 fold as the concentration of UC_22_AMPM (*C*_UC22AMPM_) increased from 0.1% to 0.5%, higher than the increment factor of UC_18_AMPM-CO_2_ aqueous foams (~1.6). These results indicated that the *C*_UC22AMPM_ exerts a more prominent influence on foamability and foam stability compared to *C*_UC18AMPM_.

The foam comprehensive index (FCI) [6], a quantitative measure to assess the foam properties, was employed to calculate the optimal concentration. The FCI can be expressed below [36]:(1)FCI=∫0t12Vdt=34Vmaxt12

As listed in Table 1, the FCI of UC_22_AMPM and UC_18_AMPM reached maximum values of 1,382,062 s·mL and 49,140 s·mL at a concentration of 0.5 wt.%, respectively. Typically, the value of FCI is greater, the foam properties are better [37]. Based on the FCI criterion, 0.5 wt.% as the optimal UC_22_AMPM and UC_18_AMPM concentration was used in the following experiments. Furthermore, we also concluded that the 0.5% UC_22_AMPM has superior foam properties to 0.5% UC_18_AMPM.

To shed light on the reasons behind the difference in properties between UC_22_AMPM and UC_18_AMPM foams at their optimum concentration, the foam evolution process, liquid content (*φ*) of aqueous foams (the ratio of the liquid volume to the foam volume) and continuous phase viscosity (*η*) of foam bulk phase were studied. As shown in Figure 2, the geometry of the bubble is spherical for both cases at the initial moment (30 s). For UC_22_AMPM foams, there was virtually no change in the bubble morphology as time progressed. In contrast, the bubbles in UC_18_AMPM aqueous foams evolved quickly into irregular polyhedral over time. At the 540th second, a substantial number of bubbles of UC_18_AMPM aqueous foams disappeared, indicative of foam bursting. In principle, the bubble shape is dependent on the *φ* of the aqueous foam [8]. In the case of high *φ* in the aqueous foams, the bubbles are uniformly spherical and densely packed. Decreasing the *φ* causes bubble deformation and the formation of defined edges. Therefore, we can conclude that the *φ* of UC_22_AMPM foams remain constant for 540 s, indicative of slow drainage. In the case of UC_18_AMPM foams, the faster bubble deformation could be interpreted by the rapid lowering of *φ*, resulting from the acceleration of the drainage process. From optical visualization, we could draw a conclusion that the foam drainage process of UC_22_AMPM foams is weaker than that of UC_18_AMPM foams.

The variation in the *φ* as a function of time is shown in Figure 3A. Evidently, the *φ* in both cases increased significantly over time during the generation process of aqueous foams, reaching a maximum liquid content (*φ*_m_) on completion of foaming. Comparatively speaking, the *φ*_m_ of the 0.5% UC_22_AMPM-CO_2_ aqueous foams was about 24.7%, greater than that of the 0.5% UC_18_AMPM-CO_2_ aqueous foams (10.6%). The lower *φ*_m_ is associated with its V_max_ (145 mL), indicative of the inferior foaming ability of UC_18_AMPM. As is well-known, the foamability is positively proportional to the C-W IFT (γ) of the surfactant solution, which can be described by using the previously reported [38]:(2)W=γA
here W and A stand for external energy applied to generate the foam and the foam area created, respectively. For a fixed W, the higher the γ is, the lower the V_max_ will be. On the basis of a previous study by Feng et al. [39], with the identical head group, the γ increases with the decrease in the hydrophobic chain length. One can conclude that the γ of the UC_18_AMPM-CO_2_ solution is higher than that of the UC_22_AMPM counterpart due to its shorter alkyl chain. Thus, the UC_18_AMPM-CO_2_ solution presents poor foamability as compared with the UC_22_AMPM counterpart.

Upon CO_2_ sparging cease, the *φ* reduced gradually with time because of the drainage. It can be seen that the UC_18_AMPM-CO_2_ aqueous foams drained in the 200s to *φ* = 0, while the *φ* of UC_22_AMPM-CO_2_ aqueous foam was 20% in this period (Figure 3A), demonstrating that the drainage from UC_18_AMPM-CO_2_ aqueous foams is faster than that of UC_22_AMPM-CO_2_ aqueous solution.

The rheological results demonstrated the UC_22_AMPM dispersion saturated with CO_2_ attained very high values of zero-shear viscosity *η_o_* (3.75 × 10^4^ mPa·s) and showed shear-thinning behavior (Figure 3B). The high magnitude of *η_o_* mirrors the presence of entangled wormlike micelles in solution [40,41]. In contrast, the *η_o_* for UC_18_AMPM samples was only ~1.0 mPa·s (Figure 3B), reflecting the absence of wormlike micelles. Numerous studies have established that drainage velocity (*V*) should vary inversely with the viscosity of the continuous phase (*η*), as the following equation [42]:(3)$$V=\frac{{\mathrm{dh}}_{f}}{\mathrm{dt}}=\frac{h_{f}^{3}}{{3\eta R}_{f}^{2}}{\Delta P}_{\mathrm{film}}$$
where ΔP_film_ stands for the difference in pressure between the film center and border, h_f_ refer to the thickness of the thin film. Using Equation (3), one can conclude that the *V* of UC_18_AMPM-CO_2_ aqueous foams is four orders of magnitude greater than that of UC_22_AMPM-CO_2_ aqueous foams, consistent with our earlier conclusion (Figure 2). The consequence of faster drainage is that the *φ* decreases rapidly, concomitant with the reduction in film thickness. The thin films tend to rupture, leading to rapid foam destruction. As a result, UC_18_AMPM-CO_2_ aqueous foams show a t_1/2_ of 445 s, which is much shorter relative to UC_22_AMPM-CO_2_ aqueous foams (9750 s) in identical conditions.

According to the aforementioned results, we attributed the differences in performance between UC_22_AMPM and UC_18_AMPM foams to their viscosity discrepancy, rooted in the different assembled structures of UC_22_AMPM and UC_18_AMPM. More specifically, UC_22_AMPM with 0.5% concentration can self-assemble into wormlike micelles, but UC_18_AMPM cannot. For the UC_22_AMPM system, the entangled worm-like micelles impart high viscosity to the foam continuous phase. During the foaming process, a large amount of liquid was transported into the foam liquid channels, forming thick foam films. The thick films would increase the thermal activation energy barrier against coalescence and Ostwald ripening. More important, the drainage is retarded by high *η*. Overall, high continuous phase viscosity retarded the three types of foam destabilization processes simultaneously, thereby enhancing the stability of foams. In contrast, the UC_18_AMPM behaved as a low *η* Newtonian fluid due to the absence of wormlike micelles, leading to the formation relatively thin foam film. Furthermore, lamellae films drained rapidly due to the low *η* of the aqueous phase. The consequence of faster drainage is that the foam film becomes thinner and prone to rupture, leading to foam destruction. Therefore, the UC_18_AMPM-CO_2_ solution presents poor foam properties as compared with the UC_22_AMPM counterpart.

### 2.2. A Comparison of the Foams Switchability

We previously demonstrated the aqueous foams stabilized by UC_22_AMPM could be turned “on” and “off” on demand through the bubbling of CO_2_ or adding NH_3_·H_2_O. It is essential to examine the switchability of the UC_18_AMPM-CO_2_ foam and to make a comparison with the UC_22_AMPM ones. The pressure and temperature are constant at 3 MPa and 80 °C, respectively, to ensure that the above two compounds can be protonated again after the neutralization of NH_3_·H_2_O.

Figure 4 depicts the parallel variations of V_max_ and t_1/2_ of both CO_2_ foam systems after the alternating addition of NH_3_·H_2_O and CO_2_. It was apparent that the t_1/2_ rose or declined accordingly with the alternative introduction of CO_2_ and NH_3_·H_2_O, suggesting the foam lifetime of both foam systems can be reversibly tuned. This finding proved that CO_2_ aqueous foams prepared from UC_18_AMPM feature switchability similar to UC22AMPM, resulting from their identical hydrophilic headgroups. As shown in Figure 2, both UC_22_AMPM and UC_18_AMPM in water can be protonated into cationic surfactants after sparging CO_2_, lowering C-W IFT by adsorbing at the CO_2_/water interface and thereby promoting foam formation. Upon NH_3_·H_2_O addition, protonated surfactant converted to a surface-inactive neutral form. Consequently, UC_22_AMPM and UC_18_AMPM would desorb from the CO_2_/water interface, disrupting the foam film and thereby leading to rapid foam destabilization.

Notably, the V_max_ of UC_22_AMPM-CO_2_ foams initially remained constant and then gradually declined as the cycle number increased (Figure 4A), demonstrating foamability weakening. By comparison, the V_max_ of UC_18_AMPM-CO_2_ foams gradually boosted as the foaming/defoaming cycle number increased (Figure 4B), indicative of enhanced foamability. On the other hand, the t_1/2_ of both CO_2_ foam systems decreased as the number of foaming/defoaming cycles increased (Figure 4A,B), indicating that foam stability deteriorated as the cycle number increased. A similar result was observed in our earlier studies, arising from the accumulation of by-products (a mixture of ammonium carbonate and bicarbonate) [21].

### 2.3. Comparison of the Effect of External Factors on Foam Properties

It has been recognized that external factors such as temperature, pressure and salinity can significantly affect the foam properties [21]. In the following subsections, the influence of these external factors on the properties of the above two CO_2_ aqueous foams was investigated comparatively using an HTHP visualization foam meter.

#### 2.3.1. Effect of Temperature

To examine the impact of temperature on the CO_2_ aqueous foams made with UC_22_AMPM or UC_18_AMPM, t_1/2_ and V_max_ were determined in a temperature range of 25–120 °C at a constant pressure of 3 MPa. As shown in Figure 5A, the V_max_ of both foams systems increased slightly with the temperature elevated, meaning that the increment of temperatures improves the foaming ability.

Compared in Figure 5B are the changes in t_1/2_ for the above two aqueous foams systems at different temperatures. Both foam systems displayed similar evolution trends, i.e., the t_1/2_ diminished steeply with the elevation of temperature, demonstrating that increased temperature would deteriorate foam stability. Many studies have revealed the elevating temperature resulted in increased C-W IFT [22] and decreased *η* [40] at constant pressure. Therefore, the foam destabilization accelerates with increasing temperature as a consequence of the higher C-W IFT and lower *η*, leading to poor foam stability.

Note also that the t_1/2_ of UC_22_AMPM-CO_2_ aqueous foams is greater than that of UC_18_AMPM-CO_2_ aqueous foams within the studied temperature scope, signifying that the CO_2_ aqueous foam stabilized by UC_22_AMPM exhibits better temperature resistance compared to UC_18_AMPM foams. In addition, the t_1/2_ of UC_22_AMPM-CO_2_ aqueous foams diminished by 5.3 fold when temperature increased from 25 to 120 °C, smaller than that of the UC_18_AMPM-CO_2_ aqueous foams (~9 fold), illustrating the impact of temperature on the stability of UC_18_AMPM-CO_2_ aqueous foams is more prevalent related to UC_22_AMPM. One explanation here could be that the P_c_ is higher than that of UC_18_AMPM due to its relatively lower *φ*.

#### 2.3.2. Effect of Pressure

As observed in Figure 6A,B, the V_max_ and t_1/2_ for both samples increased with the increasing pressure, demonstrating that increasing pressure is conducive to foaming ability and foam stability. The finding is consistent with previous studies [21,43,44] attributed to the decrease in the C−W IFT with the pressure increasing. Specifically, high pressure enhances the interactions between CO_2_ and the hydrophobic tail of surfactant molecules, reducing the contact probability between CO_2_ and water molecules and thus generating a lower C-W IFT [44]. Clearly, a lower C-W IFT enables the foam to easier form and to mitigate the foam aging process.

Interestingly, the increased scope of V_max_ of both foams showed a similar variation tendency with increasing pressure. The V_max_ for UC_22_AMPM-CO_2_ aqueous foam increased from 70 to 230 mL at the tested pressures scope; the UC_18_AMPM-CO_2_ aqueous foam increased from 54 and 150 mL under identical conditions. Their V_max_ increased by approximately three times, suggesting the effect of pressure on the foaming ability of both compounds is identical. Instead, the t_1/2_ for UC_22_AMPM-CO_2_ aqueous foam increased from 3200 and 12,400 s, showing a faint increase; while the t_1/2_ of UC_18_AMPM-CO_2_ aqueous foam underwent a slight increase from 1000 to 2200 s. The growth fold of t_1/2_ for UC_22_AMPM-CO_2_ aqueous foam is around 3.9, higher than that of UC_18_AMPM ones (2.2). These results highlighted that pressure is more prominent in enhancing the stability of UC_22_AMPM-CO_2_ aqueous foam compared with that of UC_18_AMPM-CO_2_ aqueous foam.

#### 2.3.3. Effect of Salinity

Inorganic salts have been found to modulate the surface activities [45], altering the properties of the surfactant-stabilized foam [43]. Hence, a common sodium chloride (NaCl) was used as representative inorganic salt to add the above two foam systems to clarify the effect of salt on the properties of UC_22_AMPM and UC_18_AMPM CO_2_ aqueous foams.

As depicted in Figure 7A, the V_max_ of both foams samples increased initially and then maintained constant with increasing NaCl concentration. For example, the UC_18_AMPM foam expanded from 151 and 175 mL when NaCl concentration increased from 0 to 1 wt.%; while the UC_22_AMPM foam slightly grew from 189 and 199 mL by increasing NaCl concentration from 0 to 0.5 wt.%. This means that the addition of a small amount of NaCl is beneficial for foamability. A plausible explanation could be that the addition of NaCl enhanced the adsorption of surfactant molecules at the C-W interface as a result of the charge neutralization, leading to the reducing C-W IFT, and thereby improving foaming ability [46]. Thereafter, the V_max_ of both samples remained virtually constant with a further increase in NaCl concentration. We believe that electrostatic repulsions between surfactants are sufficiently shielded at high NaCl content (≥1.0 wt.%). In this scenario, the surfactants were saturated in CO_2_/water interfaces, and the C-W IFT achieved a minimum value. Consequently, high NaCl concentrations have a negligible effect on foamability.

Compared in Figure 7B is the t_1/2_ for two cases of CO_2_ aqueous foams as a function of NaCl concentration. Overall, the t_1/2_ of the UC_22_AMPM foam samples showed a downtrend at the tested NaCl concentrations, manifesting that the addition of NaCl undermined the foam stability of UC_22_AMPM. This can be interpreted with the fact that the additional NaCl causes a transformation from linear to branched micelles, leading to a decrease in *η* [47,48]. Upon the decrease in *η*, the foam aging process would speed up, leading to rapid foam destruction. As for CO_2_ aqueous foams made from UC_18_AMPM, t_1/2_ gradually increased and then remain unchanged with increasing salinity. We also attributed this enhanced t_1/2_ to the fact that the presence of NaCl enhances the adsorption density of surfactant molecules on the CO_2_/water interface through electrostatic screening, enhancing the strength of foam lamella and therefore resisting gas diffusion between bubbles.

It is also noteworthy that the t_1/2_ of UC_22_AMPM-CO_2_ aqueous foams is higher than that of UC_18_AMPM-CO_2_ aqueous foams within the studied salinity scope, signifying that the CO_2_ aqueous foam stabilized by UC_22_AMPM exhibits better salt tolerance compared to UC_18_AMPM foams.

## 3. Materials and Methods

### 3.1. Materials

UC_18_AMPM and UC_22_AMPM were synthesized according to our previously-reported procedure [39] and confirmed by proton nuclear magnetic resonance spectroscopy (^1^H NMR, Figure 8 and Figure 9). CO_2_ (≥99.998%) was purchased from Jinnengda Gas Company (Chengdu, China) and was used as received. Sodium chloride (NaCl, 99%, GC) and NH_3_·H_2_O (25 Vol.%) were purchased from Chengdu Kelong Chemical Factory Co., Ltd. (China). CD_3_Cl (≥98% deuterium content) used for ^1^H NMR analysis was obtained from Sigma-Aldrich (Shanghai, China). The deionized water with a resistivity of 18.25 MΩ·cm used throughout this study was prepared from a quartz water purification system (UPH-I-10T, Chengdu Ultra-pure Technology Co., Ltd., Chengdu, China).

### 3.2. Preparation of Foaming Solution

A concentrated parent dispersion was prepared by adding designed amounts of surfactant samples (UC_22_AMPM or UC_18_AMPM) and deionized water to a sealed Schott-Duran bottle equipped with a magnetic bar inside. Next, the resulting mixture was stirred at 60 °C for at least 10 min, yielding low-viscosity emulsion-like dispersion. Remarkably, the dispersion concentration was calibrated by adding water to compensate for the water evaporation during the agitation process. The parent dispersions were cooled to room temperature. Then, the dispersions with desired concentration were obtained by diluting the concentrated parent dispersion with deionized water or brine.

### 3.3. Evaluation of Aqueous Foams at Atmospheric Pressure

The FoamScan setup (Figure 10, TECLIS, Lyon, France), which combines image analysis and conductivity measurements to monitor foam properties, was employed to characterize the foam properties of two types of ultra-long chain tertiary amines (i.e., UC_22_AMPM and UC_18_AMPM). Briefly, 60 mL of dispersions were placed in the glass column with a porous glass filter (pore diameter 0.2 mm) and heated to the desired temperature by an embedded electric heating system. The pressure of the chamber was fixed at atmospheric pressure. Afterward, aqueous foams were formed by bubbling CO_2_ for two minutes. The CO_2_ flow rate is constant at 100 mL/min by mass flow meters. The foam volume and liquid content were measured by five pairs of electrodes located along the glass column. The bubbles evolution was captured by the CCD (charge-coupled device) camera after the gas flow stopped.

### 3.4. Evaluation of Aqueous Foams at High Pressure

Given that the FoamScan cannot perform at high pressure, the foam properties under high-pressure conditions were evaluated by an HTHP visualization foam meter (Jiangsu Hongbo Machinery Manufacturing Co., Ltd., Haian China). A detailed description of the HTHP visualization foam meter and operating procedures have been reported in our previous work [6,7,8]. Firstly, 100 mL dispersions were pumped into the visual chamber and heated to the desired temperature by an embedded electric heating system. The CO_2_ was then bubbled into the chamber to achieve the desired pressure. Afterward, the surfactant dispersions and CO_2_ were vigorously stirred at 1100 rpm for 3 min. Once agitation ceased, the V_max_ and t_1/2_ were recorded by observing the foam height. All values were measured three times per experiment, and the average value was taken as the final result.

### 3.5. Characterization of Switchability of Aqueous Foams

To examine the switchability of aqueous foams produced from UC_22_AMPM and UC_18_AMPM, the CO_2_ and NH_3_·H_2_O (25 vol.%) were used as triggers to ‘‘switch’’ foam on and off. First, at a 3 MPa CO_2_ atmosphere, the aqueous foams were generated by the agitation of 100 mL of UC_22_AMPM and UC_18_AMPM aqueous dispersion at 1020 rpm for 3 min using an HTHP visualization foam meter, respectively. Subsequently, the appropriate amount of NH_3_·H_2_O was introduced to the CO_2_ aqueous foam system, during which the foaming and defoaming processes were tracked. This operation was repeated five times, and each cycle was separated by 10 min. All measurements were performed at 80 °C.

### 3.6. Rheological Test of Foaming Solution

The rheological measurements of the foaming solution were carried out on a Physica MCR 302 (Anton Paar, Graz, Austria) rotational rheometer equipped with a concentric cylinder geometry CC27. At atmospheric pressure, CO_2_ was first bubbled into the sample at a flow rate of 200 ± 1 mL/min for 2 min. Then, 16 mL of previously gas-treated sample was introduced to the measuring cell and thermostatically incubated at the desired temperature for 20 min prior to experimentation. A solvent trap was used to reduce water evaporation in the experiments. For all experiments, flow curves were registered in a stress-controlled mode, and the data were acquired by the software Rheoplus TM. The temperature was finely controlled by a Peltier temperature control device.

## 4. Conclusions

In this work, we investigate comparatively the properties of CO_2_ foams stabilized by UC_22_AMPM and UC_18_AMPAM and examined the evolution trend of foam properties concerning variation in external factors (i.e., temperature, pressure and salinity). The results showed that CO_2_ aqueous foams prepared from UC_18_AMPM exhibited similar switching properties to UC_22_AMPM, arising from their identical tertiary amine headgroups. However, due to the relatively long hydrophobic chain, UC_22_AMPM molecules self-assembled into wormlike micelles, but UC_18_AMPM cannot. The entanglement of these wormlike micelles into a transient network imparts high viscosity to the continuous phase of foam. During the foaming process, a large amount of liquid was transported into the foam liquid channels, forming the thicker foam film. Meanwhile, the high continuous phase viscosity of the foam system decelerates lamellae drainage. With lower drainage, the lamella remained thicker. The thicker films would enhance foam strength as well as hinder gas diffusion, arresting coalescence and Ostwald ripening, thereby enhancing the foam’s lifetime. On the contrary, the viscosity of the UC_18_AMPM sample decreased to ~1.0 mPa·s because of the absence of wormlike micelles. The lower viscosity accelerated the drainage process, weakening the strength of the foam film. The reduced strength and thickness of foam film, in turn, led to the bursting of bubbles. As a result, UC_22_AMPM foam displayed better foaming ability and foam stability compared to UC_18_AMPAM foam under identical concentrations. More importantly, for UC_22_AMPM-CO_2_ foam, the positive influence derived from pressure and concentration on its foam properties is much more pronounced than those of its UC_18_AMPM counterpart. Compared with UC_18_AMPM-CO_2_ foam, the salinity and temperature had a relatively weak negative effect on the properties of UC_22_AMPM-CO_2_ foam. In summary, this comparative study advances mechanistic insights into the role of surfactant architecture in foam properties, as well as establishes macroscopic links among foam properties, surfactant structure and environmental factors, promoting the development of such foam systems.

## Data Availability

The data presented in this study are available on request from the corresponding author.

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
