# Peer review of "A Comparative Study on CO2-Switchable Foams Stabilized by C22- or C18-Tailed Tertiary Amines"

_molecules, 2023, doi:10.3390/molecules28062567_

Round 1

Reviewer 1 Report

The paper deals with the stability and quality of foams obtained using the two  surfactants having similar structure but slight difference in hydrophobic part of the molecule, with the aim to obtain information about the possibility to correlate the structure and chemical composition, on the stability and form of the foam.

The research is clearly presented but is a bit difficult to understand the idea until the very end of the paper.

In the introduction the authors should add the importance of foam control mechanism and where are those foams useful to be applied. This is essential for the understanding the soundness of the research.

The part Materials and methods with the procedure of surfactant synthesis and foam measurement should be just after the introduction part. It would be essential to give some images of the foam that were analyzed in the research. Your work is based on the image analyzer but we don't have one image of the foam itself. Hope you can provide some images.

The separate analysis of the factors influencing the foam stability and quality is interesting and your conclusions sound well. I would like to see if the multifactorial analysis is possible, like some regression data that would incorporate the influence of parameters and give the mathematical component that is easier to follow. Like the Vmax and t1/2 depending on temperature, pressure and other parameters that you studied. Probably you can make the regression analysis of data and present in one mathematical model the influence of temperature an pressure, and than add the influence of salinity and addition of NH3. I feel that this would be useful to work with.

Reviewer 2 Report

 In This Manuscript, CO2 foam was prepared by N-isopropyl -n, n-dimethylamine (UC22AMPM) and n-oleyl - propyl -n, n-dimethylamine (UC18AMPM), two super-long tertiary amines with different tail lengths. The effects of temperature, pressure and salinity on the properties of two foam systems were compared by using a high temperature and high pressure visualization foam instrument. The continuous phase viscosity and liquid content of the two samples were characterized by rheometer and foam scanner.

The language of the article is smooth and the text is well arranged and their logic is clear. However, I think this manuscript should be revised and answer our problems. So I recommend to you that this manuscript can be accepted after revising.

1.     Why the results and discussions are arranged at the second part? This part is very important and is common set in the second part. And the results and discussion should be arranged in the third part.

2.     In the figure 7, what is the function of the Video camera in the Foam scan setup? Would you show some photographs in your manuscript?  

3.     How to control the accuracy of CO2 at a constant flow rate of 100 mL/min? Please describe it in in your manuscript.

4.     What is the reasons for the difference in performance between UC22AMPM and UC18AMPM foams at the optimal concentration? How the wormlike micelles reduce the drainage speed and increase the foam stability? Please explained it in the manuscript.

5.     Supplementary materials is short, it is commending to adding it in the manuscript.

Reviewer 3 Report

The paper positive features  are based on the very complete and deep characterization of the foam system which is certainly not easy to handle  and the  well explained use of different techniques .Certainly the results are helpful for people active in the field and the  foams propertyies correlation are  complete and accurate.

The weakness of the paper;

1,from the scientific viewpoints is first of all in the fact that a complex  correlation between molecular structure of the surfactants and foams features is developped  with only two molecules  not allowuing to give a more genarl validity to the very large amouint of work done.

2, Indeed the expalnation of the rsults is simply ascribed in the conclusion :". However, due to the relatively long hydrophobic chain, the UC22AMPM foam displayed better foaming ability and foam stability compared with UC18AMPAM foam under identical concentrations." This was easily predictable  for any having experience on hydrophylic/hydrophobic character of different  molecules,  No attempyt is offered  for understanding a more detailed mechanism of the differe4nt behaviour . Also the author have selected the  formal concentration of surfactants providing larger diferences. These last drops in going from o,5 to  o,1 % thus arising the doubt that  the solubility in water, not discussed , actually plays an important role.

Thus I would recommend  the authors to improve their contribution by considering the above comments.

Round 2

Reviewer 1 Report

I am satisfied with your revision, the importance of the work is exposed more clear than previously.

However I have some questions

Is it possible tho make diagrams in figure 1 a bit easier to read, the numbers on the image are so small that it is impossible to read.

Also, giving table 1 is the answer to the multifactorial  analysis that I would like to see. Mathematical regression is not essential at this level, but you could do it in further research.

I was pleased seeing how the manuscript improved.
